# Combined Effect of NZVI and H_2_O_2_ on the Cyanobacterium *Microcystis aeruginosa*: Performance and Mechanism

**DOI:** 10.3390/nano12173017

**Published:** 2022-08-31

**Authors:** Yun Kong, Lipeng Ji, Yue Wang, Jiake Li, Hao Lu, Shuhong Mo, Xianxun Wang, Liang Zhu, Xiangyang Xu, Xing Zheng

**Affiliations:** 1State Key Laboratory of Eco-Hydraulics in Northwest Arid Region, Xi’an University of Technology, Xi’an 710048, China; 2College of Resources and Environment, Yangtze University, Wuhan 430100, China; 3Key Laboratory of Water Pollution Control and Environmental Safety of Zhejiang Province, Hangzhou 310058, China; 4Department of Environmental Engineering, Zhejiang University, Hangzhou 310058, China

**Keywords:** *Microcystis aeruginosa*, algal organic matters, Fenton-like process, antioxidant enzyme system, fluorescence properties, removal mechanism

## Abstract

In order to eliminate the harmful cyanobacterium *Microcystis aeruginosa* and the algal organic matters (AOMs) produced by *M. aeruginosa*, the combined process of nanoscale zero-valent iron (NZVI) and hydrogen peroxide (H_2_O_2_) has been carried out, and the removal mechanism has also been clarified. As the initial cyanobacterial cell concentration is 1.0 (±0.05) × 10^5^ cells·mL^−^^1^, all the treatments of NZVI, H_2_O_2_, and NZVI/H_2_O_2_ have inhibition effects on both the Chl *a* contents and photosynthetic pigments, with the Chl *a* removal efficiency of 47.3%, 80.5%, and 90.7% on the 5th day, respectively; moreover, the variation of ζ potential is proportional to that of the Chl *a* removal efficiency. The malondialdehyde content and superoxide dismutase activity are firstly increased and ultimately decreased to mitigate the oxidative stress under all the treatments. Compared with NZVI treatment alone, the oxidation of the H_2_O_2_ and NZVI/H_2_O_2_ processes can effectively destroy the antioxidant enzyme system and then inactivate the cyanobacterial cells, which further leads to the release of photosynthetic pigments and intracellular organic matters (IOM); in addition, the IOM removal efficiency (in terms of TOC) is 61.3% and 54.1% for the H_2_O_2_ and NZVI/H_2_O_2_ processes, respectively. Although NZVI is much more effective for extracellular organic matters (EOM) removal, it is less effective for IOM removal. The results of the three-dimensional EEM fluorescence spectra analysis further confirm that both H_2_O_2_ and NZVI/H_2_O_2_ have the ability to remove fluorescent substances from EOM and IOM, due to the oxidation mechanism; while NZVI has no removal effect for the fluorescent substances from EOM, it can remove part of fluorescent substances from IOM due to the agglomeration. All the results demonstrate that the NZVI/H_2_O_2_ process is a highly effective and applicable technology for the removal of *M. aeruginosa* and AOMs.

## 1. Introduction

Control of harmful cyanobacterial blooms (HCBs) from lakes and reservoirs has become a growing concern worldwide and is always challenging for global researchers [1,2,3]. For harmful cyanobacteria, the representative toxic cyanobacterium *Microcystis aeruginosa* has received increasing attention in recent years due to the production of hazardous cyanotoxins and secretion of algal organic matters (AOMs) [4,5], along with the potential formation of disinfection by-products (DBPs) during drinking water treatment [6,7]. Many methods, including physical, chemical, and biological treatments, have been adopted for *M. aeruginosa* removal over the past few decades [8,9,10]. Although the biological treatments are efficient for removing cyanobacteria, it takes a long time [1,8]. In consequence, the simple and cost-effective ways are highly desired.

It is reported that nanoscale zero-valent iron (NZVI) has been used as a potential material for environmental pollutants remediation due to the nanoparticle size, large specific surface area, and high surface reactivity [11], and iron-based nanoparticles such as NZVI (Fe^0^), Fe_2_O_3_, and Fe_3_O_4_ are considered to have better adsorption and agglomeration effects due to the porous structure [12,13]; Recent studies have observed that the iron-based nanoparticles show both inhibition and promotion effects on the representative cyanobacterium *M. aeruginosa* [13,14,15], and the effects are dependent on the exposure dose and exposure time of the nanoparticles [16,17,18]. In order to minimize the aforementioned adverse effects, recent studies have focused on the combined process of chemical reagents.

Previous studies demonstrate that advanced oxidation processes (AOPs) can generate a great quantity of reactive radicals such as hydroxyl radicals (HO•) and sulfate radicals (SO_4_^−^•), which may show a positive effect on inactivation by damaging cyanobacterial cells [19,20]. Therefore, significant attention has been focused on the use of AOPs as cyanotoxins, and AOMs may be oxidized in the meantime [21,22]. One of the most common uses of AOPs is to catalyze the persulfate (PS) or peroxymonosulfate (PMS) with Fe (II)/Fe(III), in which the sulfate radical (SO_4_^−^•) are generated [2,6]. However, this process has some disadvantages, such as the selection of the catalyst species and sulfate accumulation in the environment [23,24]. Hydrogen peroxide (H_2_O_2_) has been considered to be an environmentally friendly and cost-effective chemical, since the products of H_2_O_2_ are H_2_O and O_2_ [25,26], while the rapid consumption of H_2_O_2_ makes it difficult to ensure the effective removal of AOMs. To overcome the above drawbacks, much attention has been focused on the Fenton and Fenton-like oxidation. For example, H_2_O_2_/Fe(II) and H_2_O_2_/Fe(III) are effective for the removal of *M. aeruginosa*, as well as microcystins [20]; and a UV/H_2_O_2_-Fe(II) process is also proven to be a highly effective technology for the *M. aeruginosa* and AOMs removal without secondary pollution [27]. 

In consideration of the Fenton-like reaction of NZVI/H_2_O_2_ showing high efficiency in degrading organic pollutants from wastewater [11], there are reasons to believe that NZVI/H_2_O_2_ is expected to be a promising and applicable technology to improve cyanobacterium removal efficiency [20]; moreover, the production of oxidizing radicals during the reaction of NZVI with H_2_O_2_ has relatively high oxidation ability and may destroy cyanobacterial cells and remove AOMs. Up to now, little research is available on using NZVI/H_2_O_2_ for *M. aeruginosa* removal through the simultaneous agglomeration and oxidation process, and the enhancement mechanism for cyanobacteria degradation by the NZVI/H_2_O_2_ process is also scarcely reported. In this study, NZVI is firstly employed as a moderate catalyst to assist H_2_O_2_ for enhancing the removal of *M. aeruginosa* and AOMs. The specific objectives of our research are as follows: (1) investigate the feasibility of employing the NZVI/H_2_O_2_ process for removing *M. aeruginosa* and AOMs; (2) evaluate the characteristics and the removal efficiency of AOMs during the treatment; and (3) reveal the mechanism of the NZVI/H_2_O_2_ process’ enhanced oxidation for *M. aeruginosa*.

## 2. Materials and Methods

### 2.1. Cyanobacterium and Chemical Reagents

The experimental *M. aeruginosa* FACHB-905 is obtained from the Freshwater Algae Culture Collection of Institute of Hydrobiology, Chinese Academy of Sciences (Wuhan, China). Before being used as inoculant, it is cultured for 7 d to reach the exponential growth phase, and the culture conditions are as follows: sterilized BG11 medium, 2000 lux white light, light/dark = 14 h/10 h, 25 ± 1 °C [8]. 

NZVI (iron powder) is purchased from Xuzhou Jiechuang New Materials Technology Co., Ltd., China (Xuzhou, China) with an average grain diameter of 100 nm and a purity of higher than 99.9%. The hydrogen peroxide used in the present study is analytical-reagent grade with a concentration of 30%.

### 2.2. Experimental Procedures

For all experiments, *M. aeruginosa* cultures in the exponential growth phase are diluted to a concentration of approximately 1.0 (±0.05) × 10^5^ cells·mL^−1^ in the 500 mL sterilized conical beakers with 250 mL BG11 medium. NZVI (50 mg·L^−1^) and/or H_2_O_2_ (5.4 mL·L^−1^) are simultaneously added into the beakers to achieve the designed concentrations, and then the beakers are brought to a final volume of 250 mL by the addition of BG11 medium (pH value of 7.0–7.2). A negative control (CK) is made by adding 1.35 mL BG11 medium into 250 mL cyanobacterial solution. The experiments are conducted in a programmable illumination incubator (GZX-III, Shanghai Xinmiao Medical Instrument Manufacturing Co., Ltd., Shanghai, China) under the aseptic condition, and the experiment conditions are set per Section 2.1. All the controls and the treatments are replicated 3 times and shaken 5 to 8 times by hand each day during the incubation. The supernatant of each sample is collected from 1~3 cm below the surface of cyanobacterial solution and is used for analysis (the beakers are sitting for more than 1 h before sampling), and the arithmetical means (±SD) are obtained and used as the final results.

### 2.3. Determination of Chlorophyll and Photosynthetic Pigments

Samples (5 mL) are filtered through a 0.45 μm GF/F filter (Whatman, UK), and the chlorophyll *a* (Chl *a*) is extracted using 10 mL of acetone (90%). The optical densities of extracts at 630, 645, 663, and 750 nm are determined using a UV-2401 PC spectrophotometer (Shimadzu, Japan) with 1 cm cell. The Chl *a* concentration is then determined according to the method described by our previous study [8]. Phycocyanobilin (PC), allophycocyanin (APC), and phycoerythrin (PE) are extracte by the freezing and thawing method, absorbencies of supernatant are determined at 565, 620, and 650 nm according to the reference [8]. The removal efficiency is calculated according to Equation (1):Removal efficiency = (1 − C_t_/C_0_) × 100%(1)
where C_0_ and C_t_ are the concentrations in the control and test groups at initial and time t, respectively.

### 2.4. Analytical Methods

#### 2.4.1. Determination of Zeta Potential

Samples (5 mL) are filtered through a 0.45 μm GF/F filter (Whatman, UK), and the filtrate is used for zeta potential determination (Li et al., 2015). The zeta potential is measured using a zeta potential analyzer (Zetasizer Nano ZS 90, Malvern, UK) [28]. 

#### 2.4.2. Determination of Antioxidant Ability

Twenty-five milliliters of each culture are collected and centrifuged for 10 min at a speed of 4000× *g*, and then the cyanobacterial cells are suspended with the phosphate buffer solution (50 mM PBS, pH 7.8) and destroyed using Ultrasonic Cell Disruption System (NingBo Scientiz Biotechnological Co., Ltd, Ningbo, China) (800 W, 5 s:5 s, 100 times) to extract enzymes. The extracting solution is centrifuged for 10 min at a speed of 10,000× *g*, and the supernatant is used for antioxidant ability analysis and intracellular organic matters (IOM) determination. The malondialdehyde level (MDA), superoxide dismutase (SOD) activity, catalase (CAT) activity, and peroxidase (POD) activity are measured according to our previous study [8].

#### 2.4.3. Total Organic Carbon Analysis

The AOMs are treated as follows, before measuring: samples (5 mL) are filtered through a 0.45 μm GF/F filter (Whatman, UK), and the filtrate is used for extracellular organic matters (EOM) determination [29]; IOM samples are obtained as mentioned in Section 2.4.2 and are filtered through a 0.45 μm GF/F membrane. The concentrations of EOM and IOM are determined as total organic carbon (TOC). TOC is measured with a TOC analyzer (TOC-2000, Metash, Shanghai, China). All measurements are conducted in triplicate, and errors are less than 2%.

#### 2.4.4. Excitation–Emission Matrix (EEM) Fluorescence Spectroscopy Analysis

The three-dimensional EEM fluorescence spectra are recorded on a F-7000 fluorescence spectrophotometer (Hitachi, Tokyo, Japan). Three-dimensional spectra are obtained by measuring the excitation wavelengths from 200 to 400 nm, and the emission spectra from 250 to 550 nm repeatedly. The excitation and emission slits are maintained at 10 nm, and the scanning speed is set at 1200 nm min^−1^ [29].

## 3. Results and Discussion

### 3.1. Growth Inhibition of NZVI and H_2_O_2_ on M. aeruginosa

The effects of NZVI and H_2_O_2_ on the growth of *M. aeruginosa* (indicated by Chl *a* contents and photosynthetic pigments) are shown in Figure 1. Results indicate that NZVI and H_2_O_2_, single or combined, have significant inhibition effects on *M. aeruginosa* during the exposure time (Figure 1a). The Chl *a* content is 186.39 ± 0.81, 98.32 ± 1.53, 36.42 ± 2.26, and 17.41 ± 0.44 μg·L^−1^ for the control, NZVI, H_2_O_2_, and NZVI/H_2_O_2_ treatments after 5 days, respectively, and the removal efficiency for the three treatments is 47.3%, 80.5% and 90.7%, respectively. The order of *M. aeruginosa* inhibition effect is NZVI < H_2_O_2_ < NZVI/H_2_O_2_. This trend is similar to that observed by Zhang et al., (2020) [20], which demonstrates that the removal efficiency of *M. aeruginosa* by single Fe(II) is lower than that of by H_2_O_2_/Fe(II).

Photosynthetic pigments are widely used for monitoring algal and cyanobacterial photosystem II (PSII) activity [8]. As shown in Figure 1b, the PC removal efficiencies are 27.1%, 37.8%, and 43.1% on the 3rd day for the treatment of NZVI, H_2_O_2_, and NZVI/H_2_O_2_, respectively, and increase to 36.1%, 40.9% and 49.2% on the 5th day, respectively, indicating the PC could be suppressed by all treatments. It is observed that the inhibition effects of PC are similar to that of the Chl *a* content. However, the results show the suppression effects of APC and PE are different from PC (Figure 1c,d). When *M. aeruginosa* is treated by H_2_O_2_ for 5 days, it shows the best APC inhibition performance among all the selected processes, and 62.6% of the APC is removed (Figure 1c). In addition, the APC removal efficiency is 41.3% and 59.0% for NZVI and NZVI/H_2_O_2_, respectively. For PE removal, oxidation treatments are much better than NZVI on the 1st day, while the removal efficiency is 75.3%, 72.5%, and 55.7% for the three treatments on the 5th day, respectively, and the APC inhibition order is NZVI > H_2_O_2_ > NZVI/H_2_O_2_ (Figure 1d), demonstrating the oxidation of H_2_O_2_ or NZVI/H_2_O_2_ on PE is not as good as the agglomeration by NZVI. The less effective removal of PE by H_2_O_2_ or NZVI/H_2_O_2_ (compared with the NZVI treatment on the 5th day) is mainly due to the decomposition of H_2_O_2_ or the consummation by the Fenton-like reaction. These results indicate that the oxidation of H_2_O_2_ or NZVI/H_2_O_2_ is good for the removal of PC and APC, while ZNVI is the best for PE removal. 

Previous studies have concluded that the microalgae removal by NZVI is size-dependent and dosage-dependent, and it is due to the agglomeration and physical interactions [12,13,30]. In addition to the agglomeration, the oxidation of H_2_O_2_ also plays an important role in the removal performance [25,31]. Moreover, the combined process of NZVI/H_2_O_2_ can greatly improve the cyanobacterium *M. aeruginosa* removal [20]. Obviously, the enhanced removal efficiency of *M. aeruginosa* by the NZVI/H_2_O_2_ process is mainly due to the oxidative effect of H_2_O_2_ and, secondly, due to the agglomeration of NZVI. The agglomeration process, which may facilitate cyanobacterial precipitation, has been investigated by many researchers [5,13,24,32]. In addition, the highly reactive NZVI containing high levels of iron oxide nanoparticles is relatively destructive to microalgae (including cyanobacteria and algae) [12,13]. 

### 3.2. Effects of NZVI and H_2_O_2_ on the Zeta Potential

Effects of NZVI and H_2_O_2_ on the zeta potential of *M. aeruginosa* are presented in Figure 2. The ζ potentials for the treatments of NZVI, H_2_O_2_, and NZVI/H_2_O_2_ are −0.17 ± 0.05, −0.07 ± 0.00, and −2.73 ± 0.14 mV on the 1st day, respectively. With the extension of exposure times, the ζ potentials for all treatments are considerably lower than the control after being exposed for 120 h, which are −6.07 ± 0.10, −4.17 ± 0.14, and −3.20 ± 1.20 mV, respectively. Interestingly, compared with the first day, the *M. aeruginosa* cells for the control have a highly negative charge on the 5th day, with a ζ potential of −0.23 ± 0.05 mV. This is most likely because more EOM is produced, and the surface properties of the cyanobacterium are changed, during the cyanobacterial growth [9]. It is observed that the variation of ζ potential is proportional to that of the Chl *a* removal efficiency, that is, the higher the removal efficiency is, the higher the ζ potential [9,28]; moreover, the ζ potential values are higher than −15.5 mV when aluminum sulphate is used as coagulant for cyanobacterial removal, while the ζ potentials are positive when there is no removal efficiency [33]. The oxidation by H_2_O_2_ is also found to affect the cell surface properties and change the membrane potential of cyanobacterial cells [25,27]. These results are consistent with the present study: that ζ potentials are negative but higher than −8 mV, and the Chl *a* removal efficiencies are increased with the increasing of ζ potentials (Figure 1a and Figure 2).

### 3.3. Effects of NZVI and H_2_O_2_ on the Antioxidant System

As reported in other studies, the addition of anticyanobacterium, iron, nanoparticles and H_2_O_2_ can generate oxidative stress and trigger antioxidant defense system responses in cyanobacteria [5,25,34,35]. Under the different treatment conditions, the antioxidant defense system responses of *M. aeruginosa* are illustrated in Figure 3. The MDA content and SOD activity induced by the H_2_O_2_ treatment are higher than those by the NZVI treatment, but lower than those by the NZVI/H_2_O_2_ treatment among the first 3 days (Figure 3a,b). It shows that the higher the *M. aeruginosa* removal efficiencies are, the higher the MDA content and SOD activity (Figure 1a,b). On the 5th day, the MDA content and SOD activity in all treatments are decreased, and the MDA contents for both control and treatments are nearly the same, while the SOD activity order is control ≈ NZVI > H_2_O_2_ > NZVI/H_2_O_2_. The above results are consistent with the prior research, that the superoxide production in cyanobacterial cells, which exhibits various strategies to mitigate the oxidative stress, is proven to be first increased and then decreased under iron or H_2_O_2_ stress [5,25].

The order of POD activity is control < NZVI < H_2_O_2_ < NZVI/H_2_O_2_ on the 1st day; whereas, with the extension of the amount of exposed time, it changes to NZVI/H_2_O_2_ < H_2_O_2_ < control < NZVI on the 5th day (Figure 3c). This phenomenon is mainly due to the excellent agglomeration ability of NZVI at the early stage of the experiment, and the oxidation ability of H_2_O_2_ or NZVI/H_2_O_2_ at the end. In comparison with POD activity, the variation of CAT activity is different (Figure 3d). Promotion of CAT activities are observed in both H_2_O_2_ and NZVI/H_2_O_2_ on the 1st day, where CAT activities reach nearly two-fold of the control level; however, the CAT activities for these two treatments show a rapid decline on the 5th day, which are only half as many as that of the control (Figure 3d). Apparently, the CAT activity for NZVI is almost the same as the control, which is because the removal mechanism of *M. aeruginosa* belongs to the agglomeration, and the cyanobacterial cell membranes are integrated during the treatment by NZVI [5,12,13,17], while they are disrupted by H_2_O_2_ or NZVI/H_2_O_2_ with the mechanism of oxidation [20,25,31]. It is suggested that the CAT activity is firstly to clear the excessive hydroxyl radicals produced by H_2_O_2_ at the early stage of the experiment [25,31] and then to eliminate the accumulation of hydroxyl radicals from the cyanobacterial cells [8,36].

Generally, both iron (high concentration) and H_2_O_2_ have inhibition effects on the growth of *M. aeruginosa* [13,15,31,37], and the production of antioxidant enzyme activities suggests that the existence of iron or H_2_O_2_ could cause cyanobacterial cells to produce reactive oxygen species (ROS), which is demonstrated by the variation of the MDA contents [5,12,25]. On the one hand, NZVI induces the agglomeration and sedimentation of *M. aeruginosa*, and the reduced photosynthetic activity leads to the accumulation of ROS and the inhibition growth of cyanobacterium; on the other hand, H_2_O_2_ or NZVI/H_2_O_2_ generate hydroxyl radicals and destroys cyanobacterial membrane systems by the oxidation, which then results in the decrease in SOD, POD, and CAT activities. Additionally, the results of the combined NZVI/H_2_O_2_ process demonstrate that NZVI agglomeration behavior promotes cyanobacterial removal, but oxidation is still the dominant function in the entire combined inhibition effect. 

### 3.4. Effects of NZVI and H_2_O_2_ on AOMs

#### 3.4.1. TOC Variations of AOMs

AOMs, including EOM and IOM produced by *M. aeruginosa*, may pose a threat to human health, especially in the drinking water treatment that disinfects with chlorine-containing disinfectant [3,7]. Exposure of *M. aeruginosa* to different treatments results in EOM and IOM variations. As Figure 4a shows, the EOM removal efficiency in terms of TOC is 37.8%, 27.6%, and 27.3% for NZVI, H_2_O_2_ and NZVI/H_2_O_2_ treatment on the 5th day, respectively, while the IOM removal efficiency is 11.7%, 61.3%, and 54.1%, respectively (Figure 4b). Compared with H_2_O_2_ and NZVI/H_2_O_2_, NZVI is much more effective for EOM removal, but it is less effective for IOM removal. The reason for this phenomenon may be that NZVI removes TOC by adsorption or agglomeration [13,30], while H_2_O_2_ and NZVI/H_2_O_2_ remove TOC by oxidation [24,25]. 

A previous study indicates that the EOM concentrations in TOC decrease from approximately 5.6 to 0.9 mg/L after being treated by H_2_O_2_/Fe(II) and H_2_O_2_/Fe(III) [20] and decrease from nearly 22.6 to 9.7 mg·L^−1^ after being treated by PMS/Fe(II) [6], which demonstrate that EOM could be removed by oxidation. Our results also show that the H_2_O_2_ and NZVI/H_2_O_2_ processes cause massive damage to the cell structure of *M. aeruginosa*, which are collaborated by the results of the photosynthetic pigments’ removal (Figure 1). Moreover, in our study, IOM could be released from the cyanobacterium and then degraded by H_2_O_2_ or NZVI/H_2_O_2_, and the removal is mainly due to the oxidation. 

#### 3.4.2. Fluorescence Properties of AOMs

Fluorescence properties have been recognized as a powerful tool for the characterization of AOMs secreted by cyanobacterium *M. aeruginosa* [28,29,30,38]. Here, variations in fluorescence properties and fluorescent intensities are observed, as shown in Figure 5 and Table 1. The fluorescent peaks after being treated by NZVI and/or H_2_O_2_ are distinctly different. For EOM fractions, there are three major fluorescent peaks in both CK and NZVI treatments: peak A represents soluble cyanobacterial metabolic byproducts with the Ex/Em of 278 nm/335 nm; peak B (Ex/Em of 314 nm/400 nm) and peak C (Ex/Em of 255 nm/406 nm) stand for fulvic-like acids (Figure 5a,b, Table 1); but only two peaks (peak B and peak C) are observed in H_2_O_2_ or NZVI/H_2_O_2_ treatments (Figure 5c,d). Moreover, the fluorescence intensities for H_2_O_2_ (NZVI/H_2_O_2_) treatment are nearly unanimous and much lower than those observed for the CK and NZVI treatments (Table 1), with removal efficiency of peak A, peak B, and peak C of 100% (100%), 38.0% (38.1%) and 55.2% (55.2%) (in terms of fluorescence intensity), which is possibly due to the oxidation mechanism; nevertheless, NZVI has no removal efficiency for fluorescent substances, as the fluorescence intensities of NZVI are almost the same as that of CK. 

For IOM fractions, there are five major fluorescent peaks in both CK and NZVI treatments: peak A (Ex/Em of 280 nm/334 nm) and peak D (Ex/Em of 282 nm/313 nm) represent soluble cyanobacterial metabolic byproducts, peak E (Ex/Em of 364 nm/451 nm) and peak F (Ex/Em of 272 nm/451 nm) represent fulvic-like acids, and peak G (Ex/Em of 232 nm/331 nm) represents aromatic proteins II (Figure 5e,f). In contrast, there is no fluorescent peak, except peak A (with low fluorescence intensity), in H_2_O_2_ or NZVI/H_2_O_2_ treatments (Figure 5g,h), which means the fluorescent substances in IOM fractions are degraded by H_2_O_2_ or NZVI/H_2_O_2_, while they are probably agglomerated by NZVI due to the decrease in fluorescent intensities (Table 1). 

AOMs from cyanobacteria are a kind of complex compound, containing polysaccharides, amino acids, proteins, peptides, organic acids, lipids, fatty acids, nucleic acids, cyanobacterial toxins, and so on [29,39]. This reveals that the fluorescence substances from AOMs can be slightly removed by nanoparticles (CuO and ZnO) [30]), and only 12.5% (in terms of fluorescence intensity) of the soluble microbial products can be coagulated by the Fe(II) [28]). These results imply that NZVI agglomeration is ineffective for removing protein-like compounds, which are strongly consisted with the present study.

However, numerous studies have shown that the combined processes of oxidation and coagulation can cause the disruption of cyanobacterial cells and the release of IOM [6,20,28]. A study concerning the effect of CaO_2_/Fe(II) on IOM indicates that the fluorescence intensities of the humic acid-like material (Ex/Em of 250~330 nm/380~375 nm) and soluble microbial products (Ex/Em of 280/330 nm) are extensively decreased with the CaO_2_ dosage increasing [28]; in another study, around 65% of the protein-like peak is declined, and no other fluorescence peaks are observed after oxidation by PMS/Fe(II) [6]; all these conclusions demonstrate that the fluorescence substances are intensively removed by oxidation. Moreover, the TOC concentrations for EOM (IOM) show similar variation trends with the fluorescence substances, which could be attributed to EOM (IOM) composition as well.

### 3.5. Discussion of the Mechanism

AOPs are used worldwide as a feasible strategy to eliminate harmful cyanobacteria [40]. Strong oxidants, such as H_2_O_2_ [25,31], CaO_2_ [28], PMS [20], chlorine, and chlorine dioxide [26] have shown a removal ability for cyanobacteria; and the combination of Fe(II) or Fe(III) with H_2_O_2_ possesses a much stronger removal efficiency, which is improved by Fe(II) or Fe(III) coagulation [20]. According to our results, as NZVI and H_2_O_2_ are simultaneously dosed into the cyanobacterial solution, it is speculated the cyanobacterial cells are firstly agglomerated by NZVI, and then Equations (2)–(6) are expected to happen as chemical oxidation processes, with Fe(III) likely formed to effectively remove cyanobacterial cells and AOMs [27]. Subsequently, the Fe(III) acts as a catalyst to produce the complex [Fe-OOH]^2+^ (Equation (7)) and HO• (Equation (8)) [20]. After the above steps, the cyanobacterial cells and AOMs are possible to be oxidized and agglomerated. It has been proven that the main removal mechanism for *M. aeruginosa* behind such procedures is attributed to the alteration of the cyanobacterial surface zeta potential [6], destruction of the antioxidant enzyme system [5], and then inactivation of the cyanobacterial cells [20]. A previous study indicates that both H_2_O_2_/Fe(II) and H_2_O_2_/Fe(III) processes could reduce the *M. aeruginosa* cells, allowing the cyanobacterial cells to settle down more easily with Fe(II) or Fe(III) [20]. 

In our study, the enhanced removal performance of cyanobacterial cells and AOMs indicates that the NZVI is converted into Fe(II) and then Fe(III), which has a more reactive surface area and higher settling property than NZVI. These results are in accordance with the scanning electron microscope images under the H_2_O_2_/Fe(III) process [20] and PMS/Fe(II) process [6], demonstrating *M. aeruginosa* could be removed efficiently by oxidation and coagulation. Based on previous studies and current results, it can be deduced that the combined effect of NZVI/H_2_O_2_ on *M. aeruginosa* cells is likely caused by the agglomeration and oxidation process. The oxidation, which more easily releases photosynthetic pigments and IOM, may be the main cause of the destruction of antioxidant enzyme system, the removal of AOMs, and the rupture of cyanobacterial cells.
2Fe^0^ + O_2_ + 4H^+^ → 2Fe^2+^ 2H_2_O(2)
2Fe^0^ + H_2_O_2_ → 2Fe^2+^ + HO^−^ + HO• + e^−^(3)
2Fe^2+^ + O_2_ + 4H^+^ → 4Fe^3+^ 2H_2_O(4)
Fe^2+^ + H_2_O_2_ → Fe^3+^ + HO^−^ + HO•(5)
Fe^3+^ + H_2_O_2_ → Fe^2+^ + HO_2_• + H^+^(6)
Fe^3+^ + H_2_O_2_ → Fe-OOH^2+^ + H^+^(7)
Fe-OOH^2+^ → Fe^2+^ + HO• + O_2_(8)

The oxidation process of NZVI/H_2_O_2_, as well as of H_2_O_2_/Fe(II) and H_2_O_2_/Fe(III) [20], has been proven to be effective for harmful cyanobacterium and AOMs removal. However, the excessive NZVI may be toxic to aquatic organisms [12,13]; moreover, the residue of cyanobacterial cells and the released cyanotoxins still exist in the aquatic environment. Therefore, the ecotoxicity of NZVI to other aquatic organisms and cyanobacteria removal in the actual eutrophic water by NZVI/H_2_O_2_ should be further considered. Notwithstanding its limitations, the present study suggests that the harmful cyanobacterium *M. aeruginosa* and AOMs could be effectively removed by the NZVI/H_2_O_2_ process.

## 4. Conclusions

The combined effect of NZVI and H_2_O_2_ on the representative toxic cyanobacterium *M. aeruginosa* is investigated in the present study. All the treatments of NZVI, H_2_O_2_, and NZVI/H_2_O_2_ have inhibition effects on both the Chl *a* contents and photosynthetic pigments, with the Chl *a* removal efficiency of 47.3%, 80.5%, and 90.7% on the 5th day, respectively; moreover, the variation of ζ potential is proportional to that of the Chl *a* removal efficiency, that is, the higher the removal efficiency is, the higher the ζ potential. The malondialdehyde content and superoxide dismutase activity are firstly increased and ultimately decreased, to mitigate the oxidative stress under all the treatments. Compared with single NZVI or H_2_O_2_, NZVI/H_2_O_2_ is much more efficient for removing *M. aeruginosa* through the simultaneous agglomeration and oxidation process. The oxidation of the H_2_O_2_ and NZVI/H_2_O_2_ processes can effectively destroy the antioxidant enzyme system and then inactivate the cyanobacterial cells, which further leads to the release of photosynthetic pigments and IOM; in addition, the IOM removal efficiency (in terms of TOC) is 61.3% and 54.1% for the H_2_O_2_ and NZVI/H_2_O_2_ processes, respectively. NZVI is much more effective for EOM removal, but it is less effective for IOM removal. The results of the three-dimensional EEM fluorescence spectra analysis further confirm that both H_2_O_2_ and NZVI/H_2_O_2_ have the ability to remove fluorescent substances from EOM and IOM due to the oxidation mechanism; while NZVI has no removal effect for fluorescent substances from EOM, it can remove part of the fluorescent substances from IOM due to the agglomeration. All the results demonstrate that the NZVI/H_2_O_2_ process is a highly effective and applicable technology for the removal of *M. aeruginosa* and AOMs.

## Figures and Tables

**Figure 1 nanomaterials-12-03017-f001:**
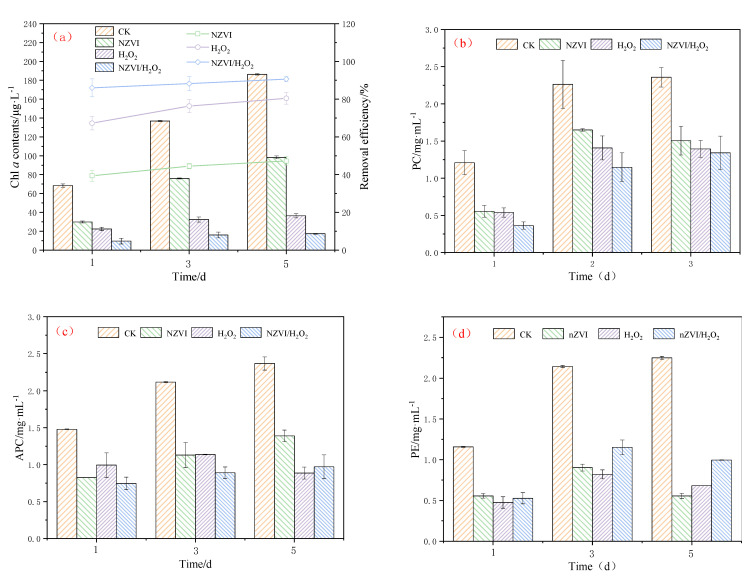
Combined effect of NZVI and H_2_O_2_ on the growth of *M. aeruginosa* for (**a**) Chl *a* contents; (**b**) PC contents; (**c**) APC contents; and (**d**) PE contents. Mean ± standard deviation of three replicates is shown for each value.

**Figure 2 nanomaterials-12-03017-f002:**
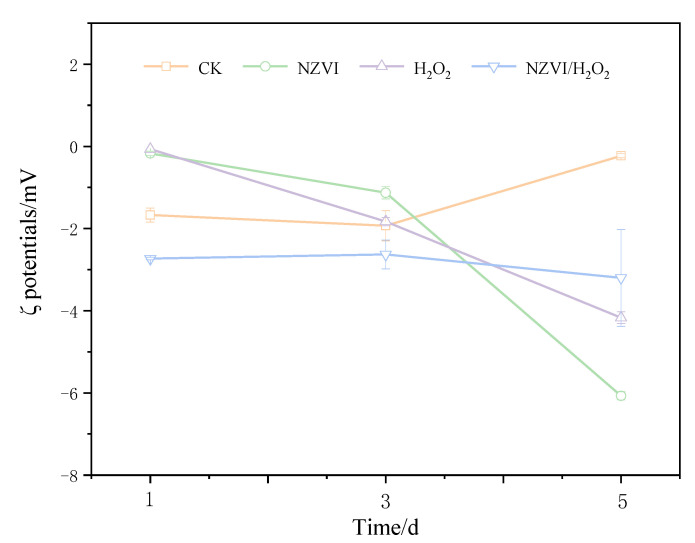
Effects of different treatments on the ζ potentials of *M. aeruginosa*. Mean ± standard deviation of three replicates is shown for each value.

**Figure 3 nanomaterials-12-03017-f003:**
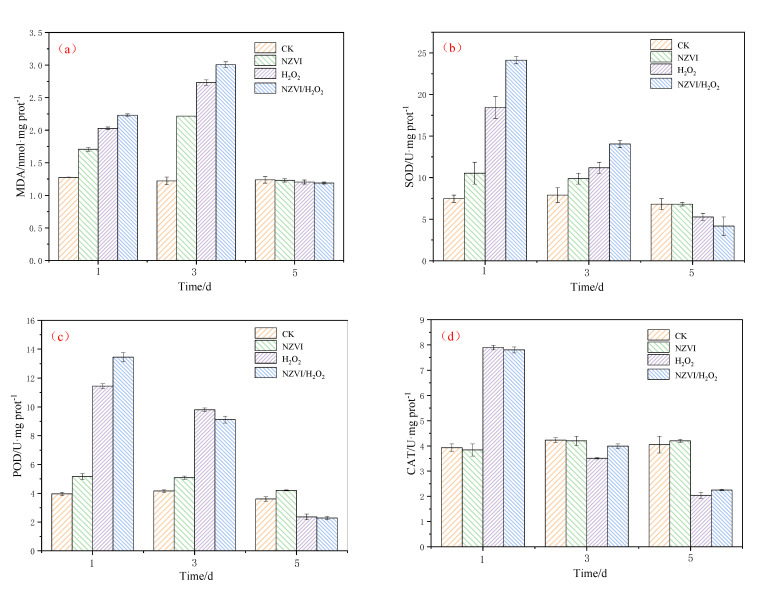
The variation of antioxidant enzyme activities in single and combined experiments for (**a**) MDA content; (**b**) SOD activity; (**c**) POD activity; and (**d**) CAT activity. Mean ± standard deviation of three replicates is shown for each value.

**Figure 4 nanomaterials-12-03017-f004:**
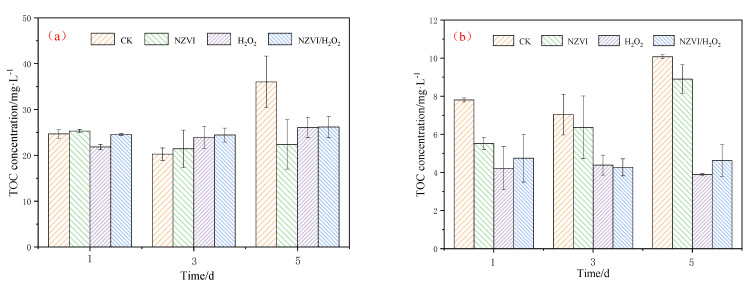
Verification of TOC for AOMs: (**a**) EOM and (**b**) IOM. Mean ± standard deviation of three replicates is shown for each value.

**Figure 5 nanomaterials-12-03017-f005:**
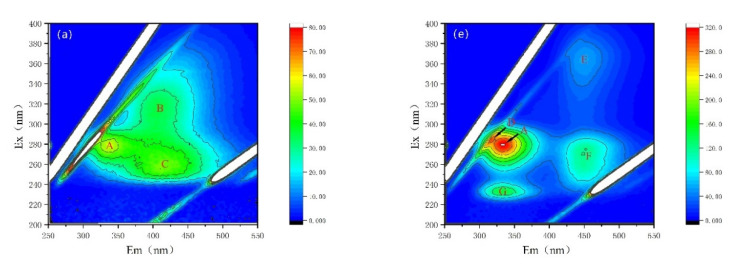
Fluorescence properties of the EOM and IOM fractions: (**a**–**d**) are the CK, NZVI, H_2_O_2_, and NZVI/H_2_O_2_ treatments for the EOM fraction; (**e**–**h**) are the CK, NZVI, H_2_O_2_, and NZVI/H_2_O_2_ treatments for the IOM fraction.

**Table 1 nanomaterials-12-03017-t001:** Fluorescence intensities of EOM and IOM fractions after 5 days exposure to different treatments.

		Peak A	Peak B	Peak C	Peak D	Peak E	Peak F	Peak G
Ex/Em(nm/nm)		278~280/334~335	314/400	255/406	282/313	364/451	272/451	232/331
Substance		Soluble cyanobacterial metabolic byproducts	Fulvic-like acids	Fulvic-like acids	Soluble cyanobacterial metabolic byproducts	Fulvic-like acids	Fulvic-like acids	Aromatic proteins II
EOM	CK	56.70	36.43	50.17	/	/	/	/
	NZVI	55.17	37.96	51.08	/	/	/	/
	H_2_O_2_	/	25.59	22.46	/	/	/	/
	NZVI/H_2_O_2_	/	25.56	22.46	/	/	/	/
IOM	CK	321.7	/	/	296.5	56.19	124.0	152.6
	NZVI	274.1	/	/	268.5	45.83	103.2	126.4
	H_2_O_2_	20.35	/	/	/	/	/	/
	NZVI/H_2_O_2_	22.82	/	/	/	/	/	/

## Data Availability

We choose to exclude this statement, since the study did not report any data.

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
