# Peer review of "Combined Effect of NZVI and H2O2 on the Cyanobacterium Microcystis aeruginosa: Performance and Mechanism"

_nanomaterials, 2022, doi:10.3390/nano12173017_

Round 1

Reviewer 1 Report

This study is an experiment on the control mechanism of cyanobacteria Microcystis aeruginosa using a mixture of NZVI and H2O2.

It is judged that the experiment or results are generally acceptable and clearly convey the purpose of the study.

However, it is very unfortunate that cyanobacteria control studies, including their previous studies, were mainly conducted indoors using cultured strains.

When cultured in a solitary state, most of the cyanobacteria cultured strains differ from natural colony-forming cyanobacteria in many ways.

Therefore, there remains a task to be finally attempted on field water or cyanobacteria in a natural state.

As a basic experiment, cyanobacteria control studies have already been conducted on the effects of numerous chemicals.

However, it is judged that transparency about problems caused by changes in water quality of field water or secondary pollution after experiments applied to field water or administration to field is more important.

Therefore, this type of research is meaningful in itself, but it is necessary to mention the side effects caused by the chemicals or the effects on the aquatic ecosystem.

In that sense, a study similar to this is equivalent to an overly simplistic test and is just another addition of data.

Rather than the direct death of cyanobacteria, additional data on the minimization of adverse effects on the space, water treatment or water purification process, and ecosystem where cyanobacteria are a problem are required.

Round 2

Reviewer 1 Report

It is judged that the authors responded sufficiently to the opinions of the reviewers. Some English correction is required for smooth English expression.

Author Response

We are grateful with the kind comments. We have asked Pro. Huaming Yao who has been worked at Georgia Institute of Technology for nearly twenty years to review the paper’s English grammar and content.  We hope that the modifications now are satisfied with the demand of the Journal.